# Host RNA Expression Signatures in Young Infants with Urinary Tract Infection: A Prospective Study

**DOI:** 10.3390/ijms25094857

**Published:** 2024-04-29

**Authors:** Kia Hee Schultz Dungu, Emma Louise Malchau Carlsen, Jonathan Peter Glenthøj, Lisbeth Samsø Schmidt, Inger Merete Jørgensen, Dina Cortes, Anja Poulsen, Nadja Hawwa Vissing, Frederik Otzen Bagger, Ulrikka Nygaard

**Affiliations:** 1Department of Pediatrics & Adolescent Medicine, Copenhagen University Hospital, Rigshospitalet, 2100 Copenhagen, Denmark; kia.hee.schultz.dungu@regionh.dk (K.H.S.D.);; 2Department of Clinical Medicine, University of Copenhagen, 2200 Copenhagen, Denmark; 3Department of Neonatology, Copenhagen University Hospital, Rigshospitalet, 2100 Copenhagen, Denmark; 4Department of Pediatrics & Adolescent Medicine, Copenhagen University Hospital North Zealand, 3400 Hillerød, Denmark; 5Department of Pediatrics & Adolescent Medicine, Copenhagen University Hospital Herlev, 2730 Herlev, Denmark; 6Department of Pediatrics & Adolescent Medicine, Copenhagen University Hospital Hvidovre, 2650 Hvidovre, Denmark; 7Center for Genomic Medicine, Copenhagen University Hospital, Rigshospitalet, 2100 Copenhagen, Denmark

**Keywords:** transcriptomics, urinary tract infection, young infants, host RNA expression signatures

## Abstract

Early diagnosis of infections in young infants remains a clinical challenge. Young infants are particularly vulnerable to infection, and it is often difficult to clinically distinguish between bacterial and viral infections. Urinary tract infection (UTI) is the most common bacterial infection in young infants, and the incidence of associated bacteremia has decreased in the recent decades. Host RNA expression signatures have shown great promise for distinguishing bacterial from viral infections in young infants. This prospective study included 121 young infants admitted to four pediatric emergency care departments in the capital region of Denmark due to symptoms of infection. We collected whole blood samples and performed differential gene expression analysis. Further, we tested the classification performance of a two-gene host RNA expression signature approaching clinical implementation. Several genes were differentially expressed between young infants with UTI without bacteremia and viral infection. However, limited immunological response was detected in UTI without bacteremia compared to a more pronounced response in viral infection. The performance of the two-gene signature was limited, especially in cases of UTI without bloodstream involvement. Our results indicate a need for further investigation and consideration of UTI in young infants before implementing host RNA expression signatures in clinical practice.

## 1. Introduction

Early diagnosis of infections in young infants remains a clinical challenge [1,2]. Young infants are particularly vulnerable to infection, and it is often difficult to clinically distinguish between bacterial and viral infections, especially in the early disease stages [2,3]. Furthermore, culture-based diagnostics take at least one day to produce a result, and established biomarkers, such as C-reactive protein and procalcitonin, have limited sensitivity [4,5]. Urinary tract infection (UTI) is the most common bacterial infection in young infants, and the incidence of associated bacteremia has decreased significantly in the recent decades [2,6,7].

Advances in molecular biology and bioinformatics have enabled omics-based approaches in the study of pathophysiology and clinical diagnostics in pediatric infectious diseases as an alternative to traditional pathogen detection methods [8]. Among these, transcriptomics presents itself as a particularly promising approach. Host RNA expression signatures have been proven capable of discriminating bacterial infections from viral infections in young infants with high sensitivity and specificity [9,10,11]. Furthermore, clinical implementation of a bedside two-gene signature is showing promising results [12]. In recent years, several countries have implemented oral antibiotic treatment for young infants with UTIs who are not suspected of bacteremia [13,14,15]. This change underscores the need for accurate early differentiation between bacterial and viral infections, as well as discerning bacterial infections with and without bloodstream involvement to minimize prolonged hospitalization and invasive procedures, as well as ensuring targeted treatment to mitigate antibiotic resistance.

Pathogen classification based on host RNA expression signatures in peripheral blood exploit altered leukocyte gene expression in response to infections and other exposures [8]. Studies have demonstrated that blood leukocytes trigger specific transcriptional responses that can be discriminated between pathogens, contributing to understanding the cellular and molecular responses that may guide targeted medical interventions [9,16,17,18]. Furthermore, potentially translating host RNA expression signatures into clinically useful biomarkers may help improve early diagnosis [10,12]. Host RNA expression signatures in peripheral blood may primarily reflect bloodstream infections, since blood serves as a migratory compartment for leukocytes [19]. However, the extent to which these signatures represent infections not involving the bloodstream remains unclarified.

In this study, we analyzed host RNA expression signatures in young infants with UTI without bacteremia compared to infants with definite viral infection. Furthermore, we tested the classification performance of a two-gene host RNA expression signature approaching clinical implementation on groups of young infants with UTI complicated by bacteremia, UTI without bacteremia, definite viral infection, probable viral infection, and non-infection.

## 2. Results

The study included 121 young infants: 7 with UTI with bacteremia, 46 with UTI without bacteremia, 33 with definite viral infection, 18 with probable viral infection, and 17 non-infected (Table 1). The uropathogens in the bacterial groups included *Escherichia coli*, *Enterococcus*, and *Enterobacter species*. The viral group included, among others, rhinovirus, enterovirus, and parainfluenza virus (Appendix A). There was a minor difference in birth weight and no significant difference in age, sex, and gestational age (Table 1).

The maximum levels of white blood cell count, absolute neutrophil count, and C-reactive protein differed between the groups, and the pairwise comparisons revealed higher levels in young infants with UTI compared to definite viral infection (Appendix A). All groups, except those non-infected, received intravenous antibiotic treatment (Table 1). Among those with UTIs with and without bacteremia, 13 of 53 (25%) had blood samples for RNA analysis collected after the initiation of treatment.

### 2.1. Host RNA Expression in UTI without Bacteremia and Definite Viral Infection

Host RNA expression analysis identified 9696 differentially expressed genes that separated young infants with UTI without bacteremia and definite viral infection. Of these, 35% were upregulated in young infants with UTI without bacteremia (Figure 1).

Unsupervised hierarchical clustering of the differentially expressed genes revealed clusters that distinguished UTI without bacteremia and definite viral infection. None of the clustering patterns were driven by gestational age, gender, and C-reactive protein (Figure 2).

### 2.2. Gene Set Enrichment Analysis

Gene set enrichment analysis revealed no significantly upregulated gene sets in young infants with UTI without bacteremia compared to definite viral infection. In contrast, we found several gene sets downregulated in young infants with UTI without bacteremia related to viral infection activity, such as “Antiviral mechanism by IFN stimulated genes”, “Interferon alpha beta signaling”, “Healthy vs. flu inf infant pbmc dn”, and “Flu vs. e coli inf pbmc up” (Table 2). Additionally, several gene sets related to innate and adaptive immune responses were also downregulated, e.g., “Complement cascade”, “Initial triggering of complement”, “Fceri mediated nfkb activation”, and “Antigen activates b cell receptor bcr leading to generation of second messengers” (Table 2).

### 2.3. Description of the Top Differentially Expressed Genes

Among the top differentially expressed genes in young infants with UTI without bacteremia, only two genes directly annotated to immune response were upregulated, i.e., *leukotriene A4 hydrolase* (*LTA4H*) and *CD44 molecule* (Figure 2, Appendix A). Three genes with uncharacterized protein product, i.e., *RP11-429G19.3*, *RP11-356C4.6*, and *RP11-67L2.2*, were among the top upregulated genes (Appendix A). In contrast, several genes were downregulated, particularly annotated to antiviral responses, e.g., *Interferon Alpha Inducible Protein 6* (*IFI6*), *negative regulator of interferon response* (*NRIR*), *ISG15 Ubiquitin Like Modifier* (*ISG15*), *MX dynamin-like GTPase 1* (*MX1*) and *2′-5′ Oligoadenylate Synthetase 1* (*OAS1*) (Figure 2, Appendix A). Several genes annotated to innate and adaptive immune response were also among the top downregulated genes, e.g., *C-C Motif Chemokine Ligand 2* (*CCL2*), *Immunoglobulin Lambda Variable 3-21* (IGLV3-21), and *Immunoglobulin Heavy Constant Gamma 3* (*IGHG3*). Genes annotated to cell proliferation, survival, migration, and differentiation included *AXL Receptor Tyrosine Kinase* (*AXL*) and *Endosome-Lysosome Associated Apoptosis and Autophay Regulator 1* (*KIAA1324*). Furthermore, several genes annotated to histone activity were downregulated, including members of the H2B, H2A, and H3 families (Figure 2, Appendix A).

### 2.4. Testing the Classification Performance of a Two-Gene Signature

The two-gene signature (genes *ADGRE1* and *IFI44L*) accurately assigned 26 of 46 young infants with UTI without bacteremia as having a bacterial infection (sensitivity 63%, 95% CI, 0.48–0.77; specificity 82%, 95% CI, 0.65–0.93, Figure 3). Further, we could not identify any alternative disease risk score threshold, as the groups had considerable overlaps in the disease risk score values. However, six of seven young infants with UTI with bacteremia were accurately assigned as having a bacterial infection (sensitivity 86%, 95% CI, 0.42–0.99; specificity 82%, 95% CI, 0.65–0.93, Figure 3).

## 3. Discussion

Our findings indicated differentially expressed genes that separated groups of young infants with UTI without bacteremia and viral infection. Only a limited number of genes annotated to the innate and adaptive immune responses were upregulated in young infants with UTI without bacteremia. In addition, three genes without annotation were also upregulated in this group. Several downregulated genes and gene sets were annotated to antiviral activity, including genes annotated to histone activity. A two-gene signature comprising *ADGRE1* and *IFI44L* demonstrated limited sensitivity for assigning young infants with UTI without bacteremia as having bacterial infection. The sensitivity of the signature for assigning young infants with UTI with bacteremia was higher but with considerable statistical uncertainty.

The limited immunological response suggested at the gene expression level in young infants with UTI without bacteremia was unexpected due to the indications of systemic inflammation, such as elevated C-reactive protein, white blood cell count, and absolute neutrophilic count, within this group. Secondly, a previous study revealed a significant overlap of more than 80% of expressed genes between peripheral blood and various tissues and organs [19]. A potential explanation for this finding is that localized leukocyte activation in the urinary tract may effectively contain the response to invading uropathogens, thus bypassing the need for universal leukocyte activation [20]. Further, gene expression changes in leukocytes recruited from the bone marrow and vessel endothelia may only be fully activated once they encounter pattern recognition receptors at the urinary tract infection site [20,21]. Upregulation of the *CD44 molecule*, known for its role in cell–cell interactions, migration, and adhesion, may align with these speculations [22]. Moreover, indication of pro-inflammatory activity was given by upregulation of *LTA4H*, a metalloenzyme crucial for the biosynthesis of leukotriene B4 [23]. Complex host responses were indicated by the genes *AXL* and *KIAA1324*. While typically known for their involvement in cell homeostasis, *AXL* has also been associated with antiviral defense mechanisms [24]. The limited immunological response contrasted with our comparison group, comprising definite viral infection, which elicited a pronounced immune response. This may reflect the fact that viruses are obligate intracellular pathogens that must infect host cells to replicate and propagate to the blood. Viral infections often induce multiple immunological mechanisms and may activate transcription of more than 100 genes, as demonstrated for interferon α/β binding to the type I interferon receptor [25]. Thus, viral infections may trigger a more enhanced widespread and universal leukocyte activation, potentially masking any immune signaling in UTI in our analysis. However, young infants with definite viral infection were considered the most clinically appropriate comparative group, given their ability to mimic bacterial infections [2]. Thus, utilizing gene expression in peripheral blood as a diagnostic tool for UTI may pose challenges and require further exploration.

Several genes without annotations were upregulated in young infants with UTI without bacteremia. This may implicate (1) previously undiscovered pathways and mechanisms in the immune responses to UTI, (2) age-specific transcription, given that the immune response in young infants differs substantially from that in older children and adults, requiring extensive adaptations that involve gene regulations [26], and (3) our use of a highly sensitive polymerase chain reaction (PCR)-free total RNA sequencing protocol recognized to increase the possibility of detecting rare genes with or without polyadenylation [27]. In addition, the notable presence of downregulated genes related to histone activity in young infants with UTI without bacteremia may indicate that epigenetic mechanisms play a role in antimicrobial defense against viral infections in this age group [28,29].

The limited performance of a previously described two-gene signature on our population has important clinical implications. While the sensitivity for correctly assigning UTI with bacteremia as bacterial infection was higher, the signature insufficiently distinguished between UTI without bacteremia and definite viral infection, the most common differential diagnoses in young infants with fever or other signs of infection. In recent years, inspiring efforts have been directed toward developing clinical diagnostic tools based on host RNA expression to discriminate bacterial from viral infections [9,10,11,12]. Our decision to focus exclusively on the two-gene signature was based on its specific development within a cohort of children with bacterial and viral infections, alongside its prior testing on a cohort of young infants [9,10]. Furthermore, the signature has progressed towards point-of-care testing, demonstrated by testing on a microchip platform [12]. Given these considerations, we considered this signature to be the most appropriate for analysis within our cohort. However, our results indicate that many young infants with UTIs would be classified incorrectly. Although unlikely to be fatal, delayed treatment of UTI is associated with the risk of spread of the infection and renal scarring [30]. Possible explanations for the limited applicability of the two-gene signature on our population may be that it was initially identified in children with an average age of 19 months and with bacterial infections in the bloodstream and cerebrospinal fluid [10]. Although the two-gene signature holds great promise as a future bedside diagnostic tool, our results highlight the importance of validation in larger populations with and without bloodstream infections before clinical implementation. Currently, host RNA expression analysis results typically require more time to obtain compared to conventional diagnostics. Furthermore, the cost and availability of RNA analysis technology present challenges. However, the field is advancing rapidly, with ongoing improvements in efficiency, cost-effectiveness, and accessibility.

Our study had some important limitations. First, the relatively small sample size of 121 young infants, with only 7 having UTI with bacteremia, limited the statistical power and the precision and generalizability of our results. Secondly, we could not validate our findings in an external cohort. In the future, our results require validation in larger cohorts and populations. However, despite these limitations, our results indicate differences in host RNA expression between young infants with UTI without bacteremia and viral infection. Furthermore, our results may have been influenced by the administration of antibiotic treatment before blood sampling in 25% of the young infants with UTI. However, our approach aligned with a previous study in which distinct host RNA expression differences were identified even when antibiotics were initiated before blood sampling [10]. Lastly, in contrast to the previous studies of host RNA expression in young infants with infections, we performed RNA sequencing instead of microarray analysis. Thus, our results are not quantitatively comparable to the results in these studies. However, RNA sequencing may offer a more unbiased approach capturing novel and rare transcripts.

## 4. Materials and Methods

This prospective study was conducted from 15 May 2020 to 31 December 2021. The study participants were young infants admitted to 4 pediatric emergency care departments in the capital region of Denmark. Inclusion criteria were age 0–89 days, born at term or late preterm, admitted from home due to symptoms of infection, and undergoing laboratory evaluation, including blood and urine cultures. Blood samples for RNA analysis were collected with clinical blood sampling at admission or as close as possible, regardless of whether antibiotics had been administered before collection. Samples collected more than 48 h after initiation of antibiotic therapy were excluded.

### 4.1. Clinical Categorization

The young infants were categorized into 5 groups: (1) UTI with bacteremia, (2) UTI without bacteremia, (3) definite viral infection, (4) probable viral infection, and (5) non-infection. UTI was defined by a positive urine culture displaying the growth of uropathogens amounting to ≥1000 colony-forming units per ml in a suprapubic aspiration or catheterization in girls. Alternatively, it was defined as ≥10,000 colony-forming units per ml of the same uropathogens in two clean catch urine samples collected within a 24 h period [31]. UTI with bacteremia was defined by detecting the uropathogen in both blood and urine cultures. Definite viral infection was defined as viral pathogen detection via PCR, e.g., nasopharyngeal swab samples. Probable viral infection was defined as symptoms of infection but no pathogen detection, maximum C-reactive protein < 50 mg/L, and negative urine culture. Non-infection was defined as young infants without fever, where infection was ruled out at the clinical evaluation.

### 4.2. Data Collection

The diagnostic work-up was performed depending on the treating pediatrician, clinical findings, and symptoms. Demographics, medical history, physical examination, laboratory results, treatment, and outcome were registered prospectively and entered into a database. A minimum of 500 μL of whole blood was collected in PAXgene blood tubes (PreAnalytiX^®^, Qiagen^®^, Hilden, Germany). A pilot study showed that 100, 250, and 500 μL of whole blood provided sufficient RNA extracted for sequencing. Gene annotations were obtained from URL: www.genecards.com (accessed on 15 December 2023).

### 4.3. RNA Sequencing, Quality Control, and Normalization

RNA was purified using QiaSymphony (Qiagen^®^, Hilden, Germany, Cat. No. 762635) and QuiaQube kits (Qiagen^®^, Hilden, Germany, Cat. No. 762174) according to the manufacturers’ instructions. Library preparation was performed using the TruSeq Stranded Total RNA Library Prep Kit (Illumina^®^, San Diego, CA, USA), and total RNA sequencing was performed on the NovaSeq 6000 (Illumina^®^) at the Center for Genomic Medicine, Copenhagen University Hospital, Rigshospitalet. After demultiplexing, the resulting per-sample FASTQ files were aligned to reference genome Hg38 using STAR version 2.5.2b [32] using parameters runThreadN 10 genomeLoad NoSharedMemory quantMode TranscriptomeSAM GeneCounts readFilesCommand zcat outSAMtype BAM SortedByCoordinate limitBAMsortRAM 35000000000, called by gnu parallel [33]. This yielded raw expression values. Quality parameters were collected at FASTQ level via FASTQC version 0.11.8. Two samples with read duplication level > 90% were excluded. Nine samples were sequenced twice to investigate library saturation, and these runs were merged at the gene count level after checking for batch effects. The batch effect was investigated from FASTQC and principal component analysis of log2 count per million gene counts.

### 4.4. Statistical Analyses

Descriptive statistics were conducted using the Kruskal–Wallis rank sum test for continuous variables and Dunn’s test for pairwise comparisons. For categorical variables, Fisher’s exact test was employed. Principal component analysis was performed for each clinical characteristic to identify potential variance drivers. The differential gene expression analysis compared UTI without bacteremia versus definite viral infection. However, to enhance the variance and robustness of our analysis, we included samples from all other groups within the study. Differential gene expression analysis with significance level α = 0.05 was performed with the package DESeq2 [34] for R statistical software, version 4.3.3. This package fits a generalized linear model to each gene based on the negative binomial distribution. Differential expression hypothesis was tested by the Wald test, and to account for multiple testing, *p*-values were adjusted using the Benjamini–Hochberg method. Based on the principal component analysis results, the model was adjusted for postnatal age. Log fold changes were shrunken by the default method [34] for use in gene set enrichment analysis [35]. Gene set names were sourced from publicly available online databases. Unsupervised hierarchical clustering on genes and samples was performed using “ward.D” clustering [36], with the Euclidean distance for the samples and Manhattan distance for the genes, respectively. To investigate the performance of a two-gene signature on all groups, a disease risk score was calculated by subtracting log2 *IFI44L* expression from log2 *ADGRE1* expression, as previously described [10,12] and plotted in a sinaplot [37]. Higher scores indicated bacterial assignment, while lower scores indicated viral assignment.

### 4.5. Study Approvals

The study was approved by the Ethics Committee of the Capital Region of Denmark (H-20028631). Informed consent was obtained from the parents of eligible young infants before participation. The study was registered at ClinicalTrials.gov (NCT04823026).

## 5. Conclusions

Our study indicated differences in host RNA expression in peripheral blood in young infants with UTI without bacteremia compared to definite viral infection. A limited immunological response was suggested in UTI without bacteremia compared to a more pronounced response in viral infection. Furthermore, the performance of a two-gene signature in distinguishing between these infections was limited, especially in cases of UTI without bloodstream involvement. Our results indicate a need for further investigation and careful consideration of UTI in young infants before implementing host RNA expression signatures in clinical practice.

## Figures and Tables

**Figure 1 ijms-25-04857-f001:**
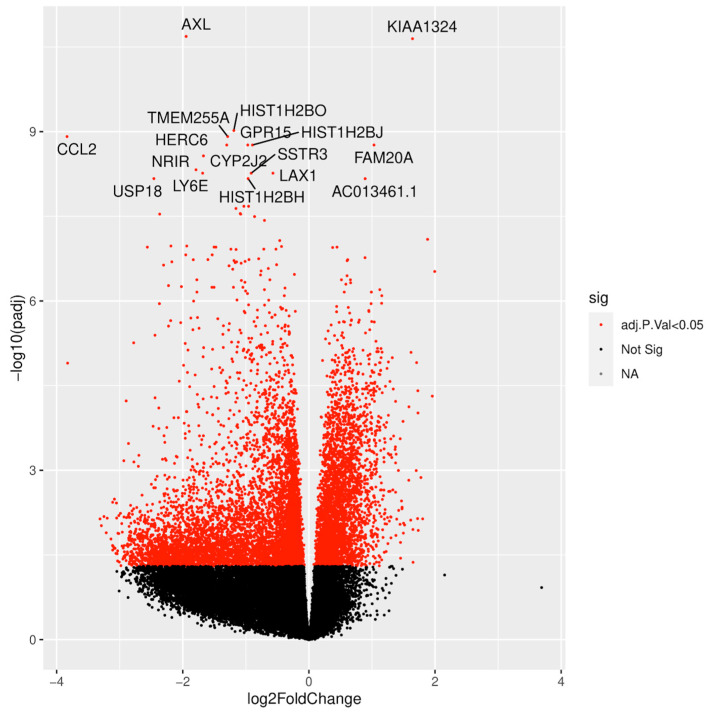
Volcano plot illustrating host RNA expression analysis comparing urinary tract infection without bacteremia and definite viral infection in young infants. The labelled genes depict the top differentially expressed genes.

**Figure 2 ijms-25-04857-f002:**
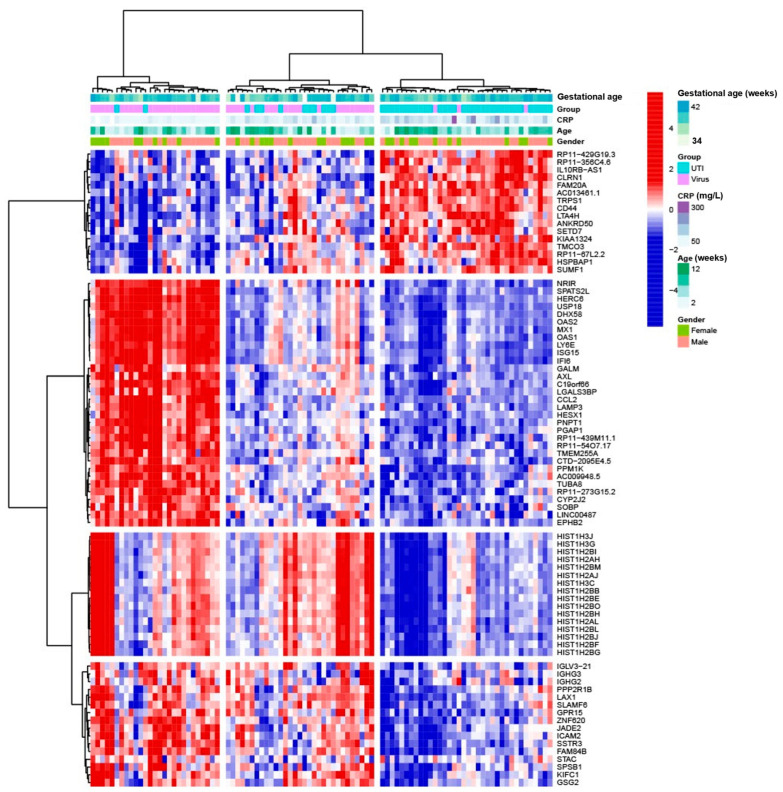
Heatmap depicting unsupervised hierarchical clustering of the top differentially expressed genes when comparing urinary tract infection without bacteremia and definite viral infection in young infants. Expression values are scaled and centered for each gene. Age is in weeks, gestational age is in weeks, and C-reactive protein is in mg/L.

**Figure 3 ijms-25-04857-f003:**
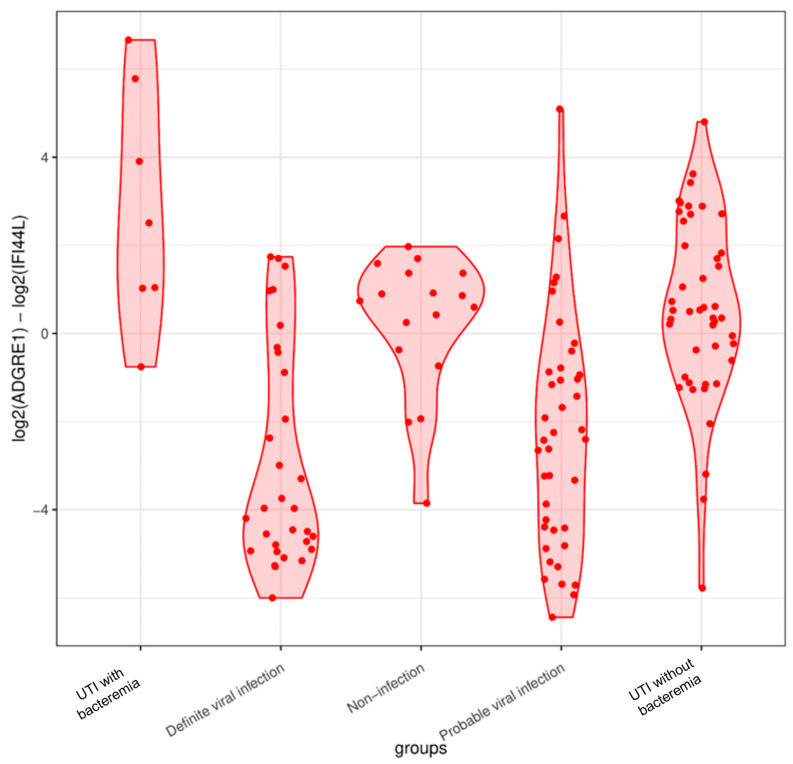
The classification performance of a 2-gene signature based on the genes *ADGRE1* and *IFI44L* for young infants with UTI with bacteremia, UTI without bacteremia, definite viral infection, probable viral infection, and non-infection. Higher disease risk scores indicated bacterial assignment and lower scores indicated viral assignment.

**Table 1 ijms-25-04857-t001:** Clinical characteristics of the study population (N = 121).

	Overall (N = 121)	UTI with Bacteremia (n = 7)	UTI without Bacteremia (n = 46)	Definite Viral Infection (n = 33)	Probable Viral Infection (n = 18)	Non-Infection (n = 17)	*p*-Value
Age (days)	38 (20–61)	19 (14–48)	54 (30–63)	37 (20–60)	44 (30–67)	25 (19–38)	0.134
Sex (male)	75 (62%)	4 (57%)	35 (76%)	15 (45%)	10 (56%)	11 (65%)	0.077
Gestational age (weeks)	39 (38–40)	38 (38–41)	39 (38–40)	39 (38–40)	39 (38–40)	38 (37–39)	0.259
Birth weight (g)	3555 (3083–3446)	3413 (3160–3446)	3710 (3450–3980)	3496 (2950–3910)	3650 (3162–3964)	3168 (2886–3350)	0.042
CRP (max)	14 (3–48)	160 (112–174)	38 (20–75)	6 (3–24)	6 (1–19)	1 (1–1)	<0.001
WBC (max)	12.2 (8.8–16.2)	15.4 (12.7–21.1)	14.4 (11.7–18.2)	9.7 (8.2–13.9)	10.9 (8.3–13.4)	7.9 (7.3–10.1)	<0.001
ANC (max)	4.6 (2.2–8.6)	8.8 (6.1–13.5)	7.0 (4.5–9.2)	3.2 (1.4–8.0)	2.5 (1.9–5.0)	2.0 (1.1–2.5)	<0.001
ALC (max)	5.3 (4.3–6.6)	5.2 (4.2–6.0)	5.6 (4.6–6.6)	5.0 (3.3–6.3)	5.7 (4.8–7.8)	4.7 (4.2–5.7)	0.301
Received antibiotics	76 (63%)	7 (100%)	46 (100%)	14 (42%)	9 (50%)	0 (0%)	<0.001

Values presented are n (%) or median (interquartile range). Statistical analyses: Kruskal–Wallis rank sum test was employed for continuous variables, followed by Dunn’s test for pairwise comparisons; Fisher’s exact test was employed for categorical variables. UTI = urinary tract infection, CRP = C-reactive protein, WBC = white blood cell, ANC = absolute neutrophil count, ALC = absolute lymphocyte count.

**Table 2 ijms-25-04857-t002:** Gene set enrichment analysis revealed several downregulated gene sets related to antiviral activity and immune response in young infants with UTI without bacteremia.

	NES	adj *p*-Value
REACTOME: Antiviral mechanism by IFN stimulated genes	↓	<0.0001
REACTOME: Complement cascade	↓	<0.0001
REACTOME: Initial triggering of complement	↓	<0.0001
REACTOME: Creation of C4 and C2 activators	↓	<0.0001
REACTOME: FCERI mediated NF KB activation	↓	<0.0001
REACTOME:	↓	<0.0001
REACTOME: FCGR3A mediated IL10 synthesis	↓	<0.0001
REACTOME: Antigen activates B cell receptor BCR leading to generation of second messengers	↓	<0.0001
REACTOME: Signaling by the B cell receptor BCR	↓	<0.0001
IGLV5 37 target genes	↓	<0.0001
GSE34205: Healthy vs. flu inf infant pbmc dn	↓	<0.0001
GSE34205: RSV vs. flu inf infant pbmc dn	↓	<0.0001
GSE6269: Flu vs. e coli inf pbmc up	↓	<0.0001

Gene set names were sourced from publicly available online databases (URL: www.gsea-msigdb.org/gsea/msigdb, accessed on 15 December 2023; URL: www.reactome.org, accessed on 15 December 2023). NES = normalized enrichment score; a downward arrow indicates a negative NES.

## Data Availability

The datasets presented in this article are not readily available, in adherence to guidelines established by the Danish Data Protection Agency, which classifies RNA sequences as personally identifiable information. Requests to access the datasets should be directed to the corresponding author.

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
