# Peer review of "Host RNA Expression Signatures in Young Infants with Urinary Tract Infection: A Prospective Study"

_ijms, 2024, doi:10.3390/ijms25094857_

Round 1
Reviewer 1 Report
Comments and Suggestions for Authors
In this paper by Schulz Dungun et al, the authors aim to define the systemic gene expression signature of infants with UTI, compared to infants with viral infections. They also test the performance of a previously described 2-gene signature for differential diagnosis. This work adds an important contribution to the understanding of the host response to UTI.
The introduction would benefit from a clearer outline of the study aims and hypothesis. A paragraph about the importance and benefit of early diagnosis, before lab tests and invasive examination, would be useful to the readers.
In the RNA expression analysis, the groups that are compared are not specified. Does the UTI group include both with and without bacteremia? Was the differential expression tested for both separately?
A fold change cutoff is commonly used to define differentially expressed genes. This should be added to the analysis.
How do the response described for infants with UTI compare to those without infection? Is the response seen here more an analysis of the absence of an anti-viral response rather than a response to a bacterial infection?
Was the hierarchical clustering tested for association with treatment? How do the authors explain the 2 different virus associated clusters?
The testing of the 2-gene signature is an important point of the study, but could be more clearly explained. Did the authors test any other gene signature, based on their study findings?
The discussion is extensive and elaborates on important point of the study. Some parts could be added to the introduction to increase the interest of the reader.
A discussion about the functional findings of the study should be discussed. What are the function and roles of the top differentially expressed genes?
Line 38: the references are not cited in sequential order.
Table 1: values are n (%). The statistical tests used should be more clearly specified.
Figure 2: units should be mentioned for the scales
Comments on the Quality of English LanguageSentences are sometimes difficult to read and would benefit from added punctuation.
Reviewer 2 Report
Comments and Suggestions for Authors
The authors present an interesting article on the detection of RNA in urinary tract infections in children, with the aim of distinguishing between viral and bacterial causes.
The article is well structured and well written, but has several limitations that need to be resolved before it can be published.
The main problem with this article is clinical applicability. Currently urine cultures are taken in children with suspected urinary tract infections to determine whether the cause is bacterial or viral. An explanation should be added about the time it takes to obtain the results of RNA in peripheral blood. In addition, there is another important issue to discuss, which is the associated cost and availability of this technique. All hospitals have a microbiology laboratory to culture urine samples, but not all have an immunology laboratory with the technology to determine RNA in peripheral blood.
In addition, there are potential false negatives due to difficulty in identifying RNA in peripheral blood.
Conclusions should be less assertive, as the results of this study are hardly extrapolable, due to all the limitations it presents.
The quality of Figure 2 needs to be improved. The rest of the figures and tables are adequate.
Comments on the Quality of English LanguageMinor editing of English language required
Round 2
Reviewer 2 Report
Comments and Suggestions for Authors
Authors have answered all the questions and requests from reviewers.
The manuscript is suitable to be publicated in this version.